# Validation of Dietary Intake Estimated by Web-Based Dietary Assessment Methods and Usability Using Dietary Records or 24-h Dietary Recalls: A Scoping Review

**DOI:** 10.3390/nu15081816

**Published:** 2023-04-08

**Authors:** Utako Murai, Ryoko Tajima, Mai Matsumoto, Yoko Sato, Saki Horie, Aya Fujiwara, Emiko Koshida, Emiko Okada, Tomoko Sumikura, Tetsuji Yokoyama, Midori Ishikawa, Kayo Kurotani, Hidemi Takimoto

**Affiliations:** 1Department of Nutritional Epidemiology and Shokuiku, National Institute of Health and Nutrition, National Institutes of Biomedical Innovation, Health and Nutrition, Kento Innovation Park NK Building, 3-17 Shinmachi, Settsu City, Osaka 566-0002, Japan; 2Department of the Science of Living, Kyoritsu Women’s Junior College, Tokyo 101-8437, Japan; 3Department of Preventive Medicine and Public Health, Keio University School of Medicine, Tokyo 160-8582, Japan; 4Department of Epidemiology and Prevention, Center for Clinical Sciences, National Center for Global Health and Medicine, Tokyo 162-8655, Japan; 5Department of Health Promotion, National Institute of Public Health, Wako 351-0197, Japan; 6Faculty of Food and Health Sciences, Showa Women’s University, Tokyo 154-8533, Japan

**Keywords:** scoping review, validity, dietary assessment

## Abstract

The goal was to summarize studies comparing the accuracy of web-based dietary assessments with those of conventional face-to-face or paper-based assessments using 24-h dietary recall or dietary record methods in the general population. Using two databases, mean differences and correlation coefficients (CCs) for intakes of energy, macronutrients, sodium, vegetables, and fruits were extracted from each study independently by the authors. We also collected information regarding usability from articles reporting this. From 17 articles included in this review, the mean dietary intake differences in the web-based dietary assessment compared to conventional methods, were −11.5–16.1% for energy, −12.1–14.9% for protein, −16.7–17.6% for fat, −10.8–8.0% for carbohydrates, −11.2–9.6% for sodium, −27.4–3.9% for vegetables, and −5.1–47.6% for fruits. The CC was 0.17–0.88 for energy, protein, fat, carbohydrates, and sodium, and 0.23–0.85 for vegetables and fruits. In three out of four studies reporting usability, more than half of the participants preferred the web-based dietary assessment. In conclusion, % difference and CC of dietary intake were acceptable in both web-based dietary records and 24-h dietary recalls. The findings from this review highlight the possibility of wide-spread application of the web-based dietary assessment in the future.

## 1. Introduction

Accurate measurements of dietary intake are fundamental for health and nutrition research. To date, the dietary record or 24-h dietary recall methods have been used for national nutrition surveys worldwide [1,2,3,4]. New technologies in dietary intake assessment, including web-based dietary assessments, have been shown to reduce issues associated with the conventional collection of dietary data (i.e., paper-based or face-to-face), such as cost, participation rates and accuracy of data collected [5,6]. Recently, the internet has become widely used [7,8], and there has been increasing interest in developing web-based dietary assessment methods [9]. Therefore, it is important to examine the possibility of applying web-based dietary assessment.

Moreover, it has become difficult to conduct in-person interviews due to the COVID-19 pandemic in 2019. In Japan, the National Health and Nutrition Survey has been annually conducted using paper-based dietary records, accompanied by in-person interviews to check information [10], but the survey was suspended in 2020 and 2021 for the first time in its history due to the pandemic [11]. On the other hand, in the UK, the national dietary survey was able to be conducted in the middle of the pandemic, since a web-based 24-h dietary recall tool was in place from 2019 [12]. Considering this situation, there is a need to examine applying web-based dietary assessment methods to the national dietary survey in Japan.

Regarding web-based 24-h dietary recalls, a previous review focused on online tools including methodology or registering foods [13]. Another review reported the validity of various mobile phone applications or web-based 24-h dietary recalls and dietary records in childhood and adolescence and concluded that those assessments may be comparably valid with respect to conventional methods [14]. However, the representative value of dietary intake was not summarized in these reviews. In dietary records of mobile phone applications, the intakes of micronutrients and food groups were underestimated compared with conventional dietary assessments in a systematic review [15]. However, that review did not systematically evaluate the validity of web-based dietary records using the internet. The validity of web-based dietary assessments has been especially limited for the elderly population [13,14,15]. The usability of web-based dietary assessment methods compared to conventional methods should be also considered, since it may affect the participation rate of the survey. Regarding usability of web-based 24-h dietary recalls, satisfaction may be considered good. On the other hand, it mentioned several comments such as difficulties in logging in or identifying correct foods in the previous review [13]. However, there was limited information about usability of web-based dietary records.

Therefore, this scoping review aimed to summarize the comparative validity of representative dietary intake estimated from web-based and conventional dietary assessment methods using dietary records or 24-h dietary recall, which are mainly used in international surveys worldwide, in the general population including children, adolescents, adults, the elderly, or those who are not used to the computer. When articles reported on usability, we collected this information.

## 2. Materials and Methods

This systematic review was planned by authors (HT, UM) with the support of authors (RT, MM) and conducted according to Preferred Reporting Items for Systematic Reviews and Meta-Analyses for Scoping Reviews (PRISMA-ScR) guidelines [16,17], as described below.

### 2.1. Search Strategy

To search the validation study of dietary intake estimated by web-based dietary assessment methods, we included words related to validation, dietary assessment, and web-based assessment methods as search terms. In addition, to examine the validity of the web-based dietary assessment methods in Japan, we included articles written in Japanese. An electronic literature search was conducted by UM on 26 February 2021 using PubMed and Web of Science without limiting the publication date. The following search terms, which were determined based on Mesh terms, were used (in English): (“food record” OR “diet record” OR “food diary” OR “dietary record” OR “recall method” OR “dietary recall” OR “diet recall” OR “24-h recall” OR “24-h recall” OR “dietary assessment”) AND (web OR internet OR automated OR mobile OR online OR digital OR “computer assisted” OR computerized) AND (validity OR validated OR validation OR comparison OR reliability) AND (English[LA] OR Japanese[LA]).

### 2.2. Inclusion Criteria

Articles eligible for inclusion: (1) were peer-reviewed original articles written in English or Japanese; (2) evaluated the validity of web-based dietary assessment compared to conventional methods (paper-based or face-to-face) as a gold standard using dietary records or 24-h dietary recalls; (3) assessed whole day intake of energy, protein, fat, carbohydrates, sodium, vegetables, and fruit; and (4) were conducted on individuals aged ≥1 years consuming a usual diet. We included studies in which parents responded on behalf of their children, and those where participants were different between web-based and conventional dietary assessments but belonged to the general population.

Articles excluded were: (1) reviews, case studies, conference reports, or abstracts; (2) focusing on specific meal occasions (e.g., breakfast); (3) using biomarkers as a gold standard; (4) focusing on patients with any diseases; (5) focusing on participants who did not eat usual diets, such as obesity, athletes, soldiers, vegetarians, and pregnant or lactating women. We included all eligible studies regardless of the study design.

This review scoped the validity of the estimated intake of energy, macronutrients, and sodium because these nutrients were important for health conditions, and vegetable and fruit intake were suggested to be important to prevent noncommunicable diseases by the World Health Organization [18]. We excluded studies using biomarkers as a gold standard because these assessment methods could not estimate representative values of dietary intake. The validity of web-based dietary assessment summarized in this study was defined as (1) an ability to estimate representative values (mean or median) of dietary intakes, and (2) an ability to rank individuals based on dietary intakes.

### 2.3. Study Selection

The titles and abstracts of all articles were screened (UM, AF, AN). After removing duplicate articles, the full texts of the articles were assessed for eligibility (UM, AF, YS, SH, EK, AN). Any disagreements were discussed and resolved by consensus or by another reviewer, if necessary (HT, RT). The reference lists of the articles identified were manually searched and screened (UM, YS, SH). The reference lists of the articles identified during this process were also examined by hand search (UM) to further identify potentially relevant articles and previous review articles [5,6,14,19,20,21,22,23,24,25,26,27,28,29,30,31]. All articles were screened or evaluated independently by at least two reviewers.

### 2.4. Data Collection

Data were extracted by two independent reviewers using an electronic spreadsheet. The information extracted included author names, publication year, study year, country, characteristics of participants (sex, age, sample size, and exclusion criteria), number of survey days, tool name, food, recipe, or dish database incorporated into the tool, and devices for web-based dietary assessment method, number of survey days of conventional dietary assessment method, mean or median intake of total energy, protein, fat, carbohydrate, sodium, and vegetables and fruit, and their correlation coefficients (CCs) both of web-based and conventional dietary assessments. When articles reported on usability such as accessibility of web-based dietary assessments, we collected this information.

### 2.5. Evaluation of the Relative Validity of Web-Based Dietary Assessment

Using the extracted information, the percentage difference of mean or median intake was calculated as the following formula: (mean or median dietary intake calculated by the web-based—mean or median dietary intake calculated by the conventional)/mean or median dietary intake calculated by the conventional × 100. According to the criteria proposed by Lombard et al. [32], the percent differences were categorized ≤10 as good; 11.0–20.0 as acceptable; >20.0 as poor, and the CCs between dietary intake estimated using the web-based and the conventional dietary assessment methods was evaluated: ≥0.50 was categorized as good; 0·20–0·49 as acceptable; <0.20 as poor. We assessed the quality score (points) of each validation study using the definition of the previous study [33]. A non-homogeneous sample (e.g., sex) was 0.5, sample size over 100 was 0.5, comparison of dietary intake was 1, correlations (0.5 for crude, 1 for energy-adjusted, and 1.5 for deattenuated or intraclass), 0.5 for agreement of dietary intake (e.g., Bland–Altman plot), 1 for data correction (face-to-face interview), 0.5 for seasonality, and 1.5 for considering supplement intake. Scores ranged from 0 to 7, and validation studies were classified as very good (>5.0), good (5.0–3.5), acceptable (3.0–2.5), and poor (<2.5). We did not conduct an adjustment/weighting of the correlation coefficient according to the quality score and a rating of the adjusted/weighted correlation, because we assessed the crude dietary intake as possible.

## 3. Results

### 3.1. Screening and Characteristics of Included Studies

Of the 856 identified articles, 562 were screened based on their titles and abstracts. Of these, 77 full-text articles were assessed for eligibility and 14 were included in this review [34,35,36,37,38,39,40,41,42,43,44,45,46,47]. In addition, we identified three studies by hand search [48,49,50] (Figure 1). Details of the included articles are presented in Table 1. All studies included this review were written in English. Out of 17 papers in this review, since the same tool was reported in two different studies, seven web-based dietary record tools and eight web-based 24-h dietary recall methods were included in this review. The included studies were conducted in nine countries: three studies in Canada [34,38,43], the UK [41,42,45], and the US [44,46,47] two in Ireland [36,40] and Brazil [49,50], and one in Belgium [48], France [37], Japan [35], and Sweden [39] (Table 1). The sample size ranged from 19 to 1147, with 10 studies conducted on <100 participants. The range of participants’ ages varied from 3 to 82 years. In addition, nine studies included participants who were recruited by email or were accustomed to the internet [34,35,36,37,38,40,41,43,45,48]. On the other hand, some studies excluded participants who were not ability to use a computer [41,44]. Except for one study conducted on nursery school children [48], participants reported their dietary intake themselves. Although the number of survey days was different between web-based and conventional methods, the most common survey period was 3 days in both methods [34,36,38,43,44,46,48,50]. The databases based on foods, recipes, and dishes were used. The range of total quality scores was 2 to 5 (Appendix A). In quality of including studies, sixteen studies were moderate (2.5 to 5 of a total of 7 points) [34,35,36,37,38,39,40,41,42,43,44,45,46,47,49,50]. the remaining one study was low (2 points) [48].

Among the seven studies which used the web-based dietary record method [35,37,38,44,46,48,49], there was only one study [44] that applied a 24-h dietary recall as the conventional method. Among the 10 studies which used the web-based 24-h dietary recall method [34,36,39,40,41,42,43,45,47,50], there were 3 studies [34,36,47] that applied the dietary record as the conventional method.

### 3.2. Difference of Dietary Intake between the Web-Based and the Conventional Dietary Assessment

Table 2 showed the mean (or median) difference in dietary intake estimated by the web-based dietary assessment method compared to those estimated by the conventional method. In web-based dietary records, the range of difference was −68.0 to 125.4 kcal (−3.1% to 6.6% difference) for energy intake, −10.4 to 3.1 g (−11.8 to 3.9% difference) for protein intake, −10.1 to 11.2g (−16.7 to 17.6% difference) for fat intake, −35.0 to 19.9 g (−10.7 to 7.8% difference) for carbohydrate intake, and 43 to 400mg (1.4 to 5.5% difference) for sodium intake. One study reported vegetable and fruit intake, and the difference was −8.0 g (−14.0% difference) for vegetable intake and 1.0 g (0.8% difference) for fruit intake. In web-based 24-h dietary recall, the range of differences was −241.0 to 342.0 kcal (−11.5% to 16.1% difference) for energy intake, −11.5 to 11.0 g (−12.1 to 14.9% difference) for protein intake, −15.0 to 10.0 g (−15.4 to 13.2% difference) for fat intake, −21.0 to 18.0 g (−7.9 to 8.0% difference) for carbohydrate intake, and −287 to 305 mg (−11.2 to 9.6% difference) for sodium intake. The range of difference was −65.0 to 3.3 g (−27.4 to 3.9% difference) for vegetable intake and −14.0 to 120.0 g for fruit intake (−5.1 to 47.6% difference).

Among the seven studies which used the web-based dietary record method [35,37,38,44,46,48,49], the one study [44] which compared to the 24-h dietary recall had the largest difference regarding fat intake (17.6%). Among the three studies [34,36,47] that compared the web-based 24-h dietary recall to the dietary record, the differences between intakes of energy, protein, fat, or carbohydrates were similar to the differences observed between web-based and conventional 24-h dietary recalls. One study [36] showed relatively large differences regarding vegetable (−27.4%) and fruits (47.6%) intakes, compared with the three studies [39,40,41] that applied the conventional 24-h dietary recall for comparisons.

### 3.3. Correlation Coefficients between the Web-Based and the Conventional Dietary Assessment

Table 3 shows the Spearman, Pearson, or Intraclass CC between dietary intake estimated by the web-based and the conventional dietary assessment methods. Overall, 12 out of 17 studies reported Spearman, Pearson, or Intraclass CC between dietary intakes. In the web-based dietary records, the CC of dietary intake was 0.37 to 0.87 for energy, 0.41 to 0.78 for protein, 0.33 to 0.75 for fat, 0.31 to 0.82 for carbohydrates, and 0.59 for sodium. There were no reports about CC of vegetable and fruit intake. In the web-based 24-h dietary recalls, the CC of dietary intake was 0.44 to 0.88 for energy, 0.41 to 0.83 for protein, 0.33 to 0.75 for fat, 0.36 to 0.82 for carbohydrate, and 0.17 and 0.75 for sodium. In food groups, there were two reported studies: the CCs were 0.23 to 0.84 for vegetable intake, and 0.56 to 0.85 for fruit intake.

Among the four studies which used the web-based dietary record method [35,38,44,49], the CCs for energy, protein, fat, or carbohydrate were low in one study [38] which applied different number of survey days for web-based and conventional methods. Among the three studies [34,36,47] that compared the web-based 24-h dietary recall to the dietary record, one study [47] showed the lowest CCs for energy, protein, carbohydrate, and sodium intakes.

### 3.4. Usability of the Web-Based Dietary Assessment Methods

Of 17 articles, 5 reported the usability of web-based dietary assessment methods [36,37,48,49], although details were not described in one study (Table 4) [44]. More than half of the participants in the three studies preferred the web-based dietary assessment [36,37,49]. However, according to Vereecken et al. [48], in the study where 58 parents were asked to either complete a web-based dietary assessment or a conventional (paper-based) dietary record, only 5 of them chose the web-based assessment. On the other hand, in the same study [48], the majority of 164 parents who were requested to answer the web-based dietary assessment indicated that the web-based dietary assessment was user-friendly (79%), attractive (68%), well-liked (66%), provided adequate instructions (92%), clear information (93%), clear pictures (91%), and easy-to-find food items (88%). Regarding registration in the dietary intake using web-based dietary assessments, participants in two of three studies answered the web-based dietary assessment was easy to report their dietary intakes on site compared with the conventional method [36,37], while a study reported that participants preferred the conventional method (paper-based) [49], and one study did not collect that information [48]. In the number of survey days, only one study reported the period for mastering the web-based dietary assessment and the most common (48.4%) answer was taking two days [37].

## 4. Discussion

This scoping review of 17 studies summarized the representative values and the CC of the intakes of energy, macronutrients, sodium, vegetables, and fruits between the web-based and conventional dietary assessments in the general population. Although the % difference in energy, protein and sodium intake was slightly smaller in web-based dietary records than in those 24-h dietary recalls, the mean difference in dietary intake estimated from both methods was within approximately ± 20% difference for energy and nutrient intakes. The CC between the web-based dietary records and conventional methods were acceptable (0.37–0.87 for energy and 0.31–0.82 for nutrients and sodium, but NA for food groups), and slightly better for web-based 24-h dietary recalls regarding energy (0.44–0.88) but not for nutrients and sodium (0.17–0.83).

We evaluated the representative dietary intakes for a whole day using web-based dietary records, and our results supported the previous study [15]. In a previous study, web-based dietary records underestimated dietary intake compared with the conventional method [15]. However, this review assessed only mobile phone applications and included studies of not assessing dietary intake for the whole day, face-to-face conventional dietary assessments, or compared with biomarkers as conventional methods [15]. In addition, we showed that the validity of dietary intake using web-based 24-h dietary recalls was also reasonable. A previous review evaluated the web-based 24-h dietary recall methods [13], and another review reported the validity of mobile phone applications or web-based dietary assessments in childhood and adolescence [14]. As this scoping review focused on the representative value of dietary intake in the general population, we excluded several studies included in previous studies [13,14]. The reasons for the excluded studies were listed in Appendix A [5,6,14,19,20,21,22,23,24,25,26,27,28,29,30,31,51,52,53,54,55,56,57,58,59,60,61,62,63,64,65,66,67,68,69,70,71,72,73,74,75,76,77,78,79,80,81,82,83,84,85,86,87,88,89,90,91,92,93,94,95,96,97,98,99,100,101,102,103].

### 4.1. Characteristics of Included Studies

In this review, only one study included participants over 70 years old [45]. In addition, included studies mainly recruited individuals who regularly used the internet or computers [34,35,36,37,40,41,43,44,45,50], although most of the included studies did not describe devices. In the national dietary survey in the UK conducted using a web-based 24-h dietary recall, participants who were not able to access through the internet or felt uncomfortable using it, or unconfident completing the dietary recalls, were supported by the survey team via telephone or internet video (Zoom) conferencing [12]. Therefore, when conducting a web-based dietary assessment for the general population including a wide range of age groups or people without internet access, it may be necessary to consider a support system. Ten studies used databases based on foods or meals [34,35,36,39,40,41,42,46,48,49]. Although two of the remaining studies used food composition tables [45,50] instead of food items as the database, five studies did not mention the database [37,38,43,44,47]. Because the database is important to estimate dietary intakes, it needed to be described in the validation studies.

As a result of this review study, the majority of studies were conducted in developed countries and no studies involving immigrants or ethnic language minority groups were included. As the number of immigrants and people living overseas increases and racial diversity expands, web-based dietary survey methods for researching these groups will also be needed. In the UK, for participants who were not able to access through the internet or felt uncomfortable using it, for example the elderly, they were helped by using devices to assist [12]. Similarly, the National Health and Nutrition Examination Survey in the US hired interpreters of various languages when necessary [104]. It is important to pay attention to the diversification of languages or emigrations when conducting web-based national dietary surveys in the future.

### 4.2. The Differences and Correlation Coefficients between Dietary Intake between the Web-Based and the Conventional Dietary Assessment

Overall, the mean or median nutrient intake calculated from the web-based dietary assessment method was underestimated compared to those from the conventional method, and it was more frequent with the web-based 24-h dietary recall method than with the web-based dietary records. Although the number of studies that described the database was limited, the number of food and beverage items was larger in the web-based dietary records than in web-based 24-h dietary recalls, and it could affect accuracy when participants registered food or beverage items.

The mean difference in dietary intake estimated by the web-based dietary assessment compared to those estimated by the conventional method was within approximately ± 20% difference for energy and nutrient intake except for fat intake in both web-based dietary records and those 24-h dietary recalls (Table 2). Regarding fat intake, sources of error between the web-based food record and the conventional method suggested that food portion estimation was the greatest source of error (50%) [44]. Another reason may be the differences in food selection, such as selecting similar but not exactly the same foods [44]. In another study [49], the database using web-based dietary assessment included 3 million items, and it seemed difficult for the subject to select the exact foods. On the other hand, Timon et al. [36] indicated a small number of databases (751 food and beverage items) as a reason for less agreement. This should be taken into account when constructing databases for web-based dietary assessments. CC of dietary intake was similar results to % difference (Table 3). Additionally, CC for sodium was lower compared to other nutrients. The web-based dietary surveys extracted in this study included dish-based dietary records [35]. In the case of the dish-based survey, the differences from the dietary records may be observed because the detailed seasonings were not checked. In addition, some systems recruited a method of selecting several portion sizes for intake [40,41]. These would have the potential to affect the estimating of the intake of salt and other seasonings.

Vegetable or fruit intake has been reported in only one study in web-based dietary records [48] and five studies in those 24-h dietary recalls. In the web-based 24-h dietary recalls, intakes of vegetables and fruits were mean different by over 20% [36], considering that dietary intakes over two different time points have been noted. CCs of vegetable and fruit intake were not reported in the web-based dietary record and only three studies were represented in those 24-h dietary recalls. Therefore, further studies are needed to confirm this association.

In the two studies, the comparison between the web-based and conventional dietary assessment methods was based on the two different populations [43,48]. Although CCs were not reported in these studies [43,48], the difference in energy intake was as small as 5% between web-based and conventional methods. As the validity of the representative values was reasonable even when the subjects of the web-based and conventional dietary assessments were different, web-based dietary assessments may be useful for a dietary assessment for the general population.

### 4.3. Usability of the Web-Based Dietary Assessment

Participants, recruited through the internet or e-mail, reported the web-based dietary assessment method made it easy to register their dietary intake on-site compared with the conventional method [36,37]. The web-based dietary assessments may be useful for people who are used to the internet which may be possible to improve the response rates of dietary assessments by using web-based dietary assessments. In a study of participants who preferred the conventional method (paper-based) [49], participants needed to register the web-based dietary record from over 3 million food items and in portion size details (e.g., grams and milliliters), which was difficult to complete, and estimate portion sizes. In the previous study, it was also noted to be difficult to identify the correct food in web-based 24-h dietary recalls [13]. According to Vereecken et al. [48], only 8% of parents who were asked to either complete a web-based or paper-based dietary assessment chose the web-based dietary assessment compared to the conventional method. The authors considered that parents did not want to spend time on the new web-based dietary assessment tools. However, in the same study [48], the majority of parents who were requested to answer the web-based dietary assessment indicated that they were favorable to the web-based method. Participants do not prefer to use the web-based dietary assessment for the first time. However, they may find it easy when they use it. Therefore, it should be considered for the recruitment and collecting of information for each generation or people who are unfamiliar with the internet. In addition, for accurate estimated dietary intake in web-based dietary assessments, careful database construction is required, such as the number of foods, dish composition, and portion size setting in both web-based dietary records and 24-h dietary recalls.

### 4.4. Strength and Limitation

The strengths of this scoping review are its investigation of the validity of representative dietary intake using the web-based dietary assessment methods compared to the conventional methods by dietary records or 24-h dietary recalls, which are used worldwide, and % difference and CC of dietary intake were reasonable in the web-based dietary records and 24-h dietary recalls. Although discussion for some nutrients is warranted, this result indicated that web-based dietary assessments may be effective as dietary assessments for the general population, especially for people who are familiar with the internet or e-mail. However, there are several limitations to this scoping review. First, despite our use of wide search terms and hand-search of reference lists, we could not capture all relevant publications. In addition, we used only two independent literature searches and searched in only two languages. Second, although we considered including the latest study, only one study included participants over 70 years old [45]. Further studies on dietary assessment methods for older people and people without familiarity with the internet are needed. Since some eligible studies were recruiting participants on the internet or who could use e-mail, we could not consider such as the participation rate of participants who do not use the internet and their usability of them. Finally, we used the criteria employed by Lombard et al. [32] to evaluate the difference between the web-based and conventional methods. This criterion also used by Luevano-Contreras et al. [105,106], is used to identify differences between two survey instruments when examining the validity and reliability of the FFQ. However, caution may need to be exercised in its interpretation, as it is not a criterion developed to compare between web-based and conventional methods.

## 5. Conclusions

The validity of web-based dietary assessment methods compared to conventional methods using dietary records and 24-h dietary recalls for dietary intake was reasonable. Although the databases and web tools used differed for most studies, the representative values of dietary intakes were consistent to some extent between the web-based and conventional dietary assessment, and it may be possible to use the web-based dietary assessment. Three of four studies reported usability, and more than half of the participants preferred the web-based dietary assessment. The results of this review provided the information needed to examine the validity of a web-based dietary evaluation method compared to conventional methods. Moreover, our results serve as both a reference and an indication for further research, as well as for the development of non-face-to-face dietary assessment methods. This review highlights findings that may be applied in designing and investigating the performance of the web-based dietary survey in the future.

## Figures and Tables

**Figure 1 nutrients-15-01816-f001:**
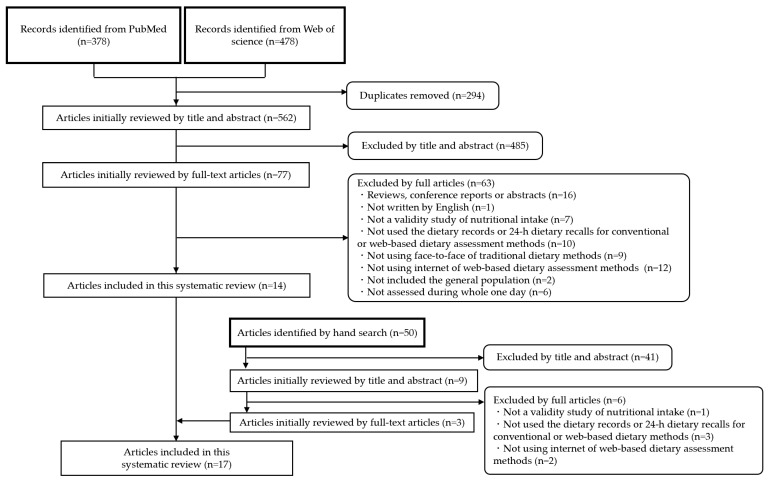
Flow diagram of the article selection process.

**Table 1 nutrients-15-01816-t001:** Characteristics of the web-based dietary records or 24-h dietary recalls.

Study	Country	Survey Year	Participants			Web-Based Dietary Assessment				Conventional Dietary Assessment	
			Sample Size (M, F)	Age, Means ± SD (Range)	Exclusion Criteria (Regarding the Internet or E-Mails)	No. of Survey Days	Tool Names	Database	Devices	Dietary Assessment Methods	No. of Survey Days
Web-based dietary records											
Matsuzaki E et al. (2017) [35]	Japan	2013–2014	163 (F: 100%)	39.3 ± 10.3	Not using e-mails	1	Internet website dish-based dietary records	Approximately 100,000 dishes	NA	Dietary Record	1
Monnerie B et al. (2015) [37]	France	2010	246 (F: 59%)	(18–60)	Not having the internet connection at home and not being accustomed to using the internet	7	MXS-Epidemio	NA	NA	Dietary Record	7
Vereecken CA et al. (2009) [48] ^a^	Belgium	2008	Web-based: 216 (NA)	3.5 ± 0.4	Not providing e-mail addresses	3	Young Children’s Nutrition Assessment on the Web	Approximately 800 different food items	NA	Dietary Record	3
			Conventional: 39 (NA)								
Raatz SK et al. (2015) [46]	US	2010–2011	19 (F: 58%)	51.6 ± 1.5	NA	3	Nutrihand	USDA-NND for Standard Reference (Release 21 as of 9/2013)	PC	Dietary Record	3
Storey KE et al. (2012) [38]	Canada	2005	459 (F: 51%, missing = 1)	12.8 (11–15)	Not contacting by e-mails	2	On-line Web-SPAN	NA	NA	Dietary Record	3
Beasley J et al. (2005) [44]	US	NA	39 (F: 54%)	53 ± 1.7 (19–69)	Lacked familiarity with personal computers	3	DietMatePro program	NA	NA	24-h Dietary Recall	1
Teixeira V et al. (2017) [49]	Brazil	NA	30 (F: 73%)	22.8 ± 2.6 (18–30)	NA	2	MyFitnessPal	More than three million food items	Smartphone, PC	Dietary Record	2
Web-based 24-h dietary recalls											
Lafrenière J et al. (2018) [34]	Canada	2015	107 (F: 53%)	47.4 ± 13.3 (18–65)	No recruited other than e-mails and e-newsletters	3	R24W	2568 food items and 687 recipes	NA	Dietary Record	3
Timon CM et al. (2017) [36]	Ireland	NA	39 (F: 51%)	32.2 ± 13.4 (18–64)	Recruited other than via e-mails, and posters	3	Foodbook24	751 food and beverage items	NA	Dietary Record	4
Lindroos AK et al. (2019) [39]	Sweden	2016–2017	78 (F: 69%)	(11–18)	NA	2	RiksmatenFlexDiet	761 core food and approximately 2300 possible food item combinations	PC, tablet, smartphone	24-h Dietary Recall	2
Timon CM et al. (2017) [40]	Ireland	2015	79 (F: 51%)	33.2 ± 12.5 (18–60)	Recruited other than by e-mails, posters, and social media and not having regular access to the internet	2	Foodbook24	751 food and beverage items	NA	24-h Dietary Recall	2
Albar SA et al. (2016) [41]	UK	NA	75 (F: 51%)	14.6 (11–18)	Having any limitation that could inhibit the adolescent’s ability to use a computer	2	Measure Your Food on One Day (myfood24)	approximately 50,000 food items	PC	24-h Dietary Recall	2
Bradley J et al. (2016) [42]	UK	2013–2014	11–16 years old: 52 (F: 63%)	(11–16)	Recruited through e-mail advertisements and snowballing techniques (aged 17–24 years).	4	INTAKE24	Over 3000 food photographs (the NDNS Nutrient Databank)	NA	24-h Dietary Recall	4
			17–24 years old: 116 F: 53%)	(17–24)							
Brassard D et al. (2020) [43] ^a^	Canada	2015–2017	Web-based (PREDISE study): 1147 (F: 50%)	(18–65)	Not having a computer, access to the internet, and a valid e-mail address	3	R24W	NA	NA	24-h Dietary Recall	1
			Conventional (CCHS): 875 (F: 50%)								
Frankenfeld CL et al. (2012) [47]	US	2010	93 (F: 65%)	27 ± 11 (18–62)	Recruited other than using flyers and web posting	2	Automated Self-Administered 24-h Dietary Records	NA	NA	Dietary Record	4
Mescoloto SB et al. (2017) [50]	Brazil	NA	40 (F: 85%)	21 (20–24)	Not owing a smartphone, people who had used the Nutrabem app before the start of the study	3	Nutrabem app	NA	NA	24-h Dietary Recall	3
Liu B et al. (2011) [45]	UK	2008	116 (F: 72%)	42 (19–82)	Recruited other than through e-mails using the mailing lists	1	Oxford WebQ	NA	NA	24-h Dietary Recall	1

SD, standard deviation; NA, not applicable; United States Department of Agriculture National Nutrient Database, USDA-NND; National Diet and Nutrition Survey, NDNS. ^a^ Participants were different for web-based and conventional dietary assessment.

**Table 2 nutrients-15-01816-t002:** Intakes of energy, macronutrient, sodium, vegetables and fruits using the web-based and conventional dietary assessment methods.

**Study**			**Energy (kcal/day)**									**Protein (g/day)**							**Fat (g/day)**				
			**Web-Based** **(Mean)**			**Conventional** **(Mean)**		**Difference**	**% Difference**			**Web-Based** **(Mean)**		**Conventional** **(Mean)**		**Difference**	**% Difference**		**Web-Based** **(Mean)**	**Conventional** **(Mean)**	**Difference**	**% Difference**	
Web-based dietary records																							
*Dietary records as the conventional method*																							
Matsuzaki E et al. (2017) [35] ^a^			1554			1472		82.0	5.6			61.3		61.6		−0.3	−0.5		45.7	45.9	−0.2	−0.4	
Monnerie B et al. (2015) [37]			1825			1836		−11.0	−0.6			75.2		77.1		−1.9	−2.5		73.2	73.8	−0.6	−0.8	
Vereecken CA et al. (2009) [48]			1294			1329		−35.0	−2.6			51.0		51.0		0.0	0.0		45.0	45.0	0.0	0.0	
Storey KE et al. (2012) [38] ^b^			2019			1893		125.4	6.6			67.9		73.0		−5.1	−6.9		71.5	68.0	3.4	5.1	
Raatz SK et al. (2015) [46]			1961			1876		85.3	4.5			82.1		79.0		3.1	3.9		79.9	77.4	2.5	3.2	
Teixeira V et al. (2017) [49]			1820 ^c^			1834 ^c^		−14.0	−0.8			77.7		88.1		−10.4	−11.8		50.2	60.3	−10.1	−16.7	
*24-h dietary recalls as the conventional method*																							
Beasley J et al. (2005) [44]			2091			2159		−68.0	−3.1			72.0		71.0		1.0	1.4		74.9	63.7	11.2	17.6	
Web-based 24-h dietary recalls																							
*24-h dietary recalls as the conventional method*																							
Lindroos AK et al. (2019) [39] ^d^			2131 ^e^			1920 ^e^		210.2	10.9			85.0		74.0		11.0	14.9		86.0	76.0	10.0	13.2	
Timon CM et al. (2017) [40] ^f^		1st	1888			2168		−241.0	−11.5			77.0		88.0		−11.0	−10.3		73.0	88.0	−15.0	−15.4	
		2nd	1817			2019		−202.0	−10.0			79.0		86.0		−7.0	−8.1		70.0	81.0	−11.0	−13.6	
Albar SA et al. (2016) [41] ^d^			1935			1989		−54.8	−2.8			68.1		70.1		−2.0	−2.8		68.3	71.3	−3.0	−4.2	
Bradley J et al. (2016) [42] ^b^		11–16 y	1597			1631		−34.0	−2.1			52.4		52.4		0.0	0.0		52.3	55.8	−3.5	−6.3	
		17–24 y	1771			1796		−25.7	−1.4			64.2		62.9		1.3	2.1		63.1	62.7	0.4	0.6	
Brassard D et al. (2020) [43]			2460			2118		342.0	16.1			-		-		-	-		-	-	-	-	
Mescoloto SB et al. (2017) [50]			1804			1950		−145.1	−7.4			88.7		86.6		2.1	2.4		65.1	76.3	−11.2	−14.7	
Liu B et al. (2011) [45]			2082 ^c^			2080 ^c^		2.6	0.1			74.3		75.3		−1.0	−1.0		79.3	75.8	3.5	5.0	
*Dietary records as the conventional method*																							
Lafrenière J et al. (2018) [34]			2595			2408		187.0	7.8			104.3		99.7		4.6	4.6		105.5	95.8	9.7	10.1	
Timon CM et al. (2017) [36]			1971			2100		−129.0	−6.1			83.5		95.0		−11.5	−12.1		78.4	85.7	−7.3	−8.5	
Frankenfeld CL et al. (2012) [47]			1831			1850		−19.0	−1.0			75.8		72.4		3.4	4.7		69.7	69.0	0.7	1.0	
**Study**		**Carbohydrate (g/day), Mean**						**Sodium (mg/day), Mean**							**Vegetables (g/day), Mean**				**Fruits (g/day), Mean**				
		**Web-Based**		**Conventional**	**Difference**		**% difference**	**Web-Based**		**Conventional**	**Difference**		**% Difference**		**Web-Based**	**Conventional**	**Difference**	**% Difference**	**Web-Based**	**Conventional**	**Difference**		**% Difference**
Web-based dietary records																							
*Dietary records as the conventional method*																							
Matsuzaki E et al. (2017) [35] ^a^		215.6		208.1	7.5		3.5	7700		7300	400		5.5		-	-	-	-	-	-	-		-
Monnerie B et al. (2015) [37]		202.0		199.0	3.0		1.5	2698		2641	57		2.2		-	-	-	-	-	-	-		-
Vereecken CA et al. (2009) [48]		171.0		180.0	−9.0		−5.0	-		-	-		-		49	57	−8.0	−14.0	125	124	1.0		0.8
Storey KE et al. (2012) [38] ^b^		273.8		253.8	19.9		7.8	-		-	-		-		-	-	-	-	-	-	-		-
Raatz SK et al. (2015) [46]		224.6		209.1	15.5		7.4	3150		3107	43		1.4		-	-	-	-	-		-		-
Teixeira V et al. (2017) [49]		207.8		232.9	−25.1		−10.8	-		-	-		-		-	-	-	-	-	-	-		-
24-h dietary recalls as the conventional method																							
Beasley J et al. (2005) [44]		292.0		327.0	−35.0		−10.7	-		-	-		-		-	-	-	-	-	-	-		-
Web-based 24-h dietary recalls																							
*24-h dietary recalls as the conventional method*																							
Lindroos AK et al. (2019) [39] ^d^		243.0		225.0	18.0		8.0	-		-	-		-		137	139	−2.0	−1.4	87	88	−1.0		−1.1
Timon CM et al. (2017) [40] ^f^	1st	226.0		247.0	−21.0		−7.9	2566		2583	−17		−4.2		142	150	−8.0	−5.3	259	273	−14.0		−5.1
	2nd	216.0		233.0	−17.0		−7.3	2168		2358	−190		−8.1		151	168	−17.0	−10.1	269	249	20		8
Albar SA et al. (2016) [41] ^d^		264.4		275.5	−11.1		−4.0	2650		2700	−50		−1.9		89	86	3.3	3.9	159	158	1.3		0.8
Bradley J et al. (2016) [42] ^b^	11–16 y	234.2		236.0	−1.8		−0.8	-		-	-		-		-	-	-	-	-	-	-		-
	17–24 y	229.1		230.3	−1.2		−0.5	-		-	-		-		-	-	-	-	-	-	-		-
Brassard D et al. (2020) [43]		-		-	-		-	3470		3165	305		9.6			-	-	-	-	-	-		-
Mescoloto SB et al. (2017) [50]		217.5		230.0	−12.5		−5.4	-		-	-		-		1.02 ^g^	1.12 ^g^	−0.1	−0.1	0.69 ^g^	0.73 ^g^	−0.04		−0.1
Liu B et al. (2011) [45]		261.9		267.3	−5.4		−2.0	-		-	-		-		-	-	-	-	-	-	-		-
*Dietary records as the conventional method*																							
Lafrenière J et al. (2018) [34]		290.6		277.7	12.9		4.6	3455		3155	301		9.5		-	-	-	-	-	-	-		-
Timon CM et al. (2017) [36]		221.0		238.0	−17.0		−7.1	2265		2552	−287		−11.2		172	237	−65.0	−27.4	372	252	120.0		47.6
Frankenfeld CL et al. (2012) [47]		233.0		233.0	0.0		0.0	3340		3500	−160		−4.6		-	-	-	-	-	-	-		-

^a^ Dietary intakes were median intake. ^b^ Dietary intakes were mean value of two times of web-based and three times of conventional dietary assessments. ^c^ Energy intake (kcal) calculated by KJ × 0.239. ^d^ Dietary intakes were mean value in 2 days both of web-based and conventional dietary assessments. ^e^ Energy intake (kcal) calculated by MJ × 238.846. ^f^ Dietary intakes were collected a web-based and conventional dietary assessment on two separate occasions over a one-month period. ^g^ Portion size.

**Table 3 nutrients-15-01816-t003:** Correlation coefficients for intakes of energy, macronutrient, vegetable, and fruit between web-based and conventional dietary assessment methods.

		Correlation Coefficients						
		Energy	Protein	Fat	Carbohydrate	Sodium	Vegetable	Fruit
Web-based dietary records								
*Dietary records as the conventional method*								
Matsuzaki E et al. (2017) [35] ^a^		0.87	0.78	0.75	0.82	0.59	-	-
Storey KE et al. (2012) [38] ^b^		0.37	0.41	0.33	0.31	-	-	-
Teixeira V et al. (2017) [49] ^a^		0.67	0.53	0.59	0.58	-	-	-
*24-h dietary recalls as the conventional method*								
Beasley J et al. (2005) [44]		0.71	0.62	0.51	0.80	-	-	-
Web-based 24-h dietary recalls								
*24-h dietary recalls as the conventional method*								
Lindroos AK et al. (2019) [39] ^b, e^		0.53	0.57	0.57	-	-	0.23	0.56
Timon CM et al. (2017) [40]	1st	0.62	0.77	0.75	0.65	0.75	0.84 ^a^	0.76 ^a^
	2nd	0.72	0.79	0.61	0.80	0.63	0.84 ^a^	0.85 ^a^
Albar SA et al. (2016) [41] ^d, e^		0.88	0.77	0.75	0.81	0.46	0.47	0.67
Mescoloto SB et al. (2017) [50]		0.77	0.83	0.71	0.82	-	0.43 ^f^	0.78 ^f^
Liu B et al. (2011) [45] ^a^		0.58	0.59	0.57	0.66	-	-	-
*Dietary records as the conventional method*								
Lafrenière J et al. (2018) [34]		0.57	0.61	0.54	0.53	0.55	-	-
Timon CM et al. (2017) [36] ^c^		0.54	0.75	0.33	0.53	0.30	-	-
Frankenfeld CL et al. (2012) [47]		0.44	0.41	0.46	0.36	0.17	-	-

Data were not available from Monnerie B et al. (2015) [37], Vereecken CA et al. (2009) [48], Raatz SK et al. [46], Bradley J et al. (2016) [42] and Brassard D et al. (2020) [43]. ^a^ Spearman coefficient correlation. Otherwise, Pearson’s correlation coefficient. ^b^ Mean value of dietary intake in 2 days of web-based and 3 days of conventional dietary assessment. ^c^ Energy-adjusted dietary intake. ^d^ Intraclass correlation coefficients. ^e^ Mean value of dietary intake in 2 days both of web-based and conventional dietary assessments. ^f^ Portion size.

**Table 4 nutrients-15-01816-t004:** Summary of usability of the web-based dietary assessment methods.

Web-Based Dietary Assessments	
Negative Points	Positive Points
Difficult to record Difficult in estimating food portions Pressure to use	Did not change the usual diet User-friendliness (perception of the data collection tool was conventional, quick and easy, and appreciation of the general layout of the questionnaire) Preferred method Practicality Self-consciousness of food habits Time required to respond to the dietary assessment was short Provided adequate instructions provided clear information, clear pictures, and easy-to-find food items

Summarized by the authors, from descriptions in Timon CM et al. [36], Monnerie B et al. [37], Vereecken CA et al. [48], Teixeira V et al. [49].

## Data Availability

Not applicable.

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
