# Peer review of "Validation of Dietary Intake Estimated by Web-Based Dietary Assessment Methods and Usability Using Dietary Records or 24-h Dietary Recalls: A Scoping Review"

_nutrients, 2023, doi:10.3390/nu15081816_

Round 1

Reviewer 1 Report

This study is an interesting review of studies on the association between web-based dietary assessment and either conventional dietary records or 24-hour dietary recalls. Please refer to the following list for a summary of the points of concern. 

L90–94: I understand that the terms regarding the dietary assessment are consistent with the terms in the title of the article referenced in the introduction. Please explain and discuss how the terms regarding web-based were chosen and whether there were enough. 

L147–148: It would be helpful to have a specific explanation of the correlations score in the text, such as how many points the energy-adjusted correlations are. 

L185–198: Some studies compared web-based dietary records with conventional 24-hour dietary recalls and others compared web-based 24-hour dietary recalls with conventional dietary records. If this study aimed to compare web-based and conventional methods, please also explain and discuss the characteristics of the results of studies which different dietary assessment methods. 

L279–312: It would be better if you also discuss why the correlation coefficients for sodium are different in each study compared to other nutrients. 

[Minor] 

L51, Table 1, L266: I think the expressions United Kingdom and UK should be unified. 

L150: Isn't the maximum score 6.5 points instead of 7? 

L151: Which would a study with 3.5 points be classified as "good" or "acceptable"? 

L163–164: Isn't the number of web-based 24-hour dietary recall methods 8 instead of 10? If my understanding is different, please supplement regarding the same tool in two different studies. 

L176: There are two "." in this sentence. 

Table 1: The hyphen in "Ref-rence" is necessary? "NDNS" should be spelled out. 

Footnotes to Table 1: The order of full spellings and abbreviations is inconsistent. Looks like "USDA-FCD" is not on Table 1. 

L198: Are these values for fruit intake? 

L203: The "T" seems to be a double-byte character. 

Table 3: At the table head states "Pearson correlation coefficients", is that accurate?

L222: The order of the references is inconsistent.

L294–295: Please provide a reference.

Reviewer 2 Report

Current paper compared web-based dietary assessment with paper base method for validation of dietary intake. This research is realistic especially for Covid-19 pandemic as well as post pandemic period with the medical survey questionnaire digitalize. However, I have several questions as below:

 Question 1: Current study included articles written in Japanese, from Pubmed and web of science (Japanese language paper in these two international databases may be rear)?

Please add information about this part. As well as the detail that how many papers in English, and how many in Japanese.

 Question 2: If included Japanese papers, the author used Japanese key words to search?

 Question 3: Page 3 line 140. The evaluation was based on criteria proposed by Lombard et al. reference 35. Cause this kind of qualitative research may do not have a golden cutoff, so is this criterion often used? If so, please also add previous papers also used this criterion.

 Question 4: Page 10 line 220-235

3.4. Usability of the web-based dietary assessment methods

This part, author summarized different opinion of web-based dietary assessment methods compared with paper base. If there is a graph or table, summarize positive thinking and negative thinking, maybe easier to grasp the points.

Reviewer 3 Report

This great systematic review shows the applicability and practicality of online tools to assess food intake, which is often overlooked in medicine. A meta-analysis was omitted due to the heterogeneity of the studies.

I have one remark, though. Even in developed countries where people have full access to ICTs, there often exists a digital divide, typically disadvantaging certain groups (women, elderly, ethnic language minorities, migrants,...), which automatically excludes the use of this rather simple tool in public health. The authors are encouraged to add a paragraph to the Discussion, explaining how they would approach this issue.

Round 2

Reviewer 1 Report

Thank you for your response.

The authors added the results of the studies compared web-based dietary records with conventional 24-hour dietary recalls and the studies compared web-based 24-hour dietary recalls with conventional dietary records. I think this review comparing web-based and conventional methods also needs to include a discussion of the comparison of the results from studies using the same and different dietary assessments (i.e., food records and 24-hour dietary recalls).

[Minor]

Footnotes to Table 1: Again, I point this out. The order of full spellings and abbreviations is inconsistent (i.e., some order “abbreviation, full spelling,” others “full spelling, abbreviation.”)

L256: Isn't the reference [39-42, 48] instead of [39-41, 42]?